# Expression and Transcript Localization of *star*, *sf-1,* and *dax-1* in the Early Brain of the Orange-Spotted Grouper *Epinephelus coioides*

**DOI:** 10.3390/ijms23052614

**Published:** 2022-02-27

**Authors:** Ganesan Nagarajan, Adimoolam Aruna, Yousef Ahmed Alkhamis, Roshmon Thomas Mathew, Ching-Fong Chang

**Affiliations:** 1Basic Sciences Department, PYD, King Faisal University, Hofuf-420, Al-Asha 31982, Saudi Arabia; 2Center of Excellence for the Ocean, National Taiwan Ocean University, Keelung 20224, Taiwan; 3Department of Aquaculture, National Taiwan Ocean University, Keelung 20224, Taiwan; n.aruna@yahoo.com; 4Animal and Fish Production Department, College of Agricultural and Food Sciences, King Faisal University, Hofuf-420, Al-Asha 31982, Saudi Arabia; yalkhamis@kfu.edu.sa; 5Fish Resources Research Center, King Faisal University, Hofuf-420, Al-Asha 31982, Saudi Arabia; rmathew@kfu.edu.sa

**Keywords:** steroidogenic enzyme gene, *sf-1*, *dax-1*, early brain development, sex differentiation, fish

## Abstract

We investigated the developmental expression and localization of *sf-1* and *dax-1* transcripts in the brain of the juvenile orange-spotted grouper in response to steroidogenic enzyme gene at various developmental ages in relation to gonadal sex differentiation. The *sf-1* transcripts were significantly higher from 110-dah (day after hatching) and gradually increased up to 150-dah. The *dax-1* mRNA, on the other hand, showed a decreased expression during this period, in contrast to *sf-1* expression. At the same time, the early brain had increased levels of steroidogenic gene (*star*). *sf-1* and *star* hybridization signals were found to be increased in the ventromedial hypothalamus at 110-dah; however, *dax-1* mRNA signals decreased in the early brain toward 150-dah. Furthermore, the exogenous estradiol upregulated *star* and *sf-1* transcripts in the early brain of the grouper. These findings suggest that *sf-1* and *dax-1* may have an antagonistic expression pattern in the early brain during gonadal sex differentiation. Increased expression of steroidogenic gene together with *sf-1* during gonadal differentiation strongly suggests that *sf-1* may play an important role in the juvenile grouper brain steroidogenesis and brain development.

## 1. Introduction

Steroid synthesis is a biochemical process that occurs in the nervous system as well as the classical peripheral steroidogenic organs, notably in the brain [1]. These neurosteroids participate in a variety of physiological and behavioral processes during the early developmental period that are still unclear. The transcriptional regulation of neurosteroidogenic enzymes in the brain has been reported in a number of teleosts in relation to gonadal differentiation [2,3,4,5,6,7] and adult neurogenesis [8]. These findings provide evidence that neurosteroid production is under the control of brain steroidogenic enzymes. However, the expressive role of steroidogenic factors and their localization on the early brain steroidogenesis during gonadal differentiation remains unknown. In the steroidogenic pathway, the steroidogenic acute regulatory protein (Star) has been shown to be important in the regulation of cholesterol, which is transported from the outer to the inner mitochondrial membrane [9] and this is the rate-limiting step in the biosynthesis of steroids since *cyp11a1*, which metabolizes cholesterol, is located at the inner mitochondrial membrane. Characterization and molecular expression of *star* have been reported in different vertebrates during development, which includes chicken [10], rat and marmoset [11], and teleosts [2,12,13,14,15]. Despite this, little is known about the developmental expression and transcript localization of steroidogenic genes in relation to orphan nuclear receptors (*sf-1* and *dax-1*) in the developing brain.

*SF-1* (*Steroidogenic Factor-1, Ad4BP, NR5a1*) is an orphan member of the nuclear hormone receptor superfamily and a homolog of the drosophila fushi tarazu factor-1 protein (*FTZ-F1*) [16,17]. *SF-1*/*sf-1* plays a key role in the development and function of steroidogenic tissues within the hypothalamic–pituitary–gonadal axis and as an essential factor in sex differentiation of many species [3,18,19,20,21,22,23,24,25,26,27]. It was originally identified in mammals as a tissue-specific transcriptional regulator of the cytochrome P450 hydroxylases, including *cyp19a1a* [28]. Since then, *SF-1*/*sf-1* has been found to influence the transcription of several steroidogenic genes in human, adrenal, and gonadal tissues, including *star* [29], *cyp11a1* [30], *3**β-hsd* [31], and *cyp19a1a* [32,33,34]. It remains unclear whether there is a cooperative mechanism of *sf-1* regulation on *cyp19a1b* expression in the early brain development of the sex-changing fish. However, promoter motif analysis in air-breathing catfish (*Clarias gariepinus*) indicated that *sf-1* binds to *cis*-acting regions in the upstream region of *cyp19a1b*, implying that *sf-1* gene regulates *cyp19a1b* transcription via binding to the promoter of *cyp19a1b* in the brain [35]. Therefore, the majority of findings show that *sf-1* is a critical element in the development and sexual differentiation of steroidogenic tissues, particularly the gonad.

Another orphan nuclear receptor is *dax-1* (dosage-sensitive sex reversal, adrenal hypoplasia congenita, critical region on the X-chromosome, gene 1). It functions as a negative co-regulator of many genes involved in the steroidogenic pathway [36,37,38,39,40]. *dax-1* has been linked to female gonadal sex differentiation in chicken [41] and pig [42]. *DAX-1* suppresses *CYP19* expression in mice by binding to its promoter regions [43]. Furthermore, it has been reported that *dax-1* inhibits *sf-1* and *foxl2*-mediated *cyp19a1a* expression in medaka ovarian follicles [36], whereas *SF-1* regulates the transcription of the *DAX-1* gene in adult rats [44]. The role of *dax-1* as a crucial factor for organ and sex development has been well documented in zebrafish (*Danio rerio*) [45,46,47,48]. In comparison to *sf-1*, there is little known about how *dax-1* might regulate potential target genes during the early developmental period. It needs to be addressed for the precise role of *sf-1* and *dax-1* in the regulation of neurosteroidogenic enzyme genes in the early development of the brain and temporal correlation to gonadal sex differentiation. Therefore, we hypothesize that *sf-1* and *dax-1* play important roles in the steroidogenesis in the early brain.

Orange-spotted grouper (*Epinephelus coioides*), an economically important prime aquaculture species and a popular marine food fish widely cultured in Taiwan, Saudi Arabia, and southeast Asian countries. Grouper is a protogynous hermaphrodite species with several unique characteristic features that make them particular interest to study gene expression profiles that are crucial for early brain development and gonadal differentiation. The fish are sex differentiated to females when they are around 4–5-month-old juvenile, and then maintain for several years before changing to male [5,6]. Thus, this mono-female sex fish provides a suitable model to study early brain development [49]. The mRNA expression of *cyp19a1a* and *sf-1* in the gonad of orange-spotted grouper after 17α-methyltestosterone-induced precocious sex transition has been observed [50]. However, the developmental expression pattern of *sf-1* and *dax-1* in relation to other key steroidogenic enzyme genes, as well as its possible expressive role in the developing brain during gonadal differentiation, remain unknown. Therefore, we investigated the temporal expression profile of *sf-1* and *dax-1* in relation to the other key steroidogenic gene in the early ages of the grouper brain during gonadal differentiation. We used *star* as a key marker of the first step in the cascade of estrogen synthesis. Further, we localized the developmental transcript expression of *star*, *sf-1,* and *dax-1* at 90-, 110-, and 150-dah (days after hatching) in the hypothalamic regions. Finally, we analyzed the effect of exogenous estradiol (E_2_) on the mRNA expression of *star, sf-1*, and *dax-1* in the grouper brain.

## 2. Results

### 2.1. Isolation of Grouper Brain star, sf-1 and dax-1

The genes *star* (GU929702), *sf-1* (JQ320496) and *dax-1* (GU929703) were partially cloned from the cDNA of the orange-spotted grouper brain. The CLUSTAL X program, version 1.81 (Conway Institute UCD, Dublin, Ireland) was used with a default setting to align the deduced amino acid sequence of genes *star*, *sf-1,* and *dax-1* from the orange-spotted grouper and other species. The partial sequence of the grouper genes showed higher identities with other teleost species: *star* gene shared 93.9%, 93.9%, 93.1%, 92.4%, 90.8%, and 77.1% identities with Japanese sea bass (JQ995529), Japanese amberjack (LC102214), gilthead seabream (EF640987), largemouth bass (DQ166820), Atlantic croaker (DQ646787), and zebrafish (AF220435), respectively; *sf-1* gene shared 97.4%, 97.4%, 97.3%, 96.9%, 96.4%, and 82.2% identities with Japanese flounder (JX999939), European seabass (JQ755267), largemouth bass (XM038715638), black porgy (AY491378), Odontesthes (DQ441595), and zebrafish (BC163522), respectively. The *dax-1* gene shares 97.4%, 95.5%, 94.2%, 93.5%, and 67.2% with black porgy (EF423617), European seabass (AJ633646), swamp eel (KF770791), spotted scat (JQ740597), and zebrafish (NM001082947), respectively (Appendix A).

### 2.2. q-PCR Analysis of sf-1 and dax-1 Transcripts in the Brain of Orange-Spotted Grouper during Developmental Ages

#### 2.2.1. *sf-1* mRNA Expression

The neuroanatomy section of the early brain is shown in Figure 1. In the telencephalon, mesencephalon, and diencephalon, *sf-1* mRNA levels gradually increased from 110- to 150-dah, although transcript levels were found to be low at 180-dah (Figure 2A–C). In the telencephalon, mesencephalon, and diencephalon, *sf-1* mRNA expression was significantly (*p* < 0.05) higher from 110- to 150-dah compared to 90-dah. Furthermore, when compared to the telencephalon and mesencephalon, the level of mRNA in the diencephalon was higher (Figure 2A–C). The levels of *sf-1* mRNA in the diencephalon and mesencephalon at 150-dah were 2.5- and 1.8-fold greater, respectively, than the levels at 90-dah (Figure 2A–C).

#### 2.2.2. *dax-1* mRNA Expression

Unlike *sf-1* expression, *dax-1* mRNA expression was significantly (*p* < 0.05) higher at 90-dah and subsequently declined from 110- to 150-dah in the telencephalon and mesencephalon, and 110- to 130-dah in the diencephalon (Figure 2D–F). In 90-dah, the amounts of *dax-1* transcripts in the telencephalon, mesencephalon, and diencephalon increased significantly (*p* < 0.05). When compared to the levels in the telencephalon at 90-dah, higher expression of *dax-1* in the mesencephalon and diencephalon was 2.2- and 1.6-fold higher, respectively (Figure 2D–F). Furthermore, higher expression of *dax-1* mRNA was identified in the mesencephalon at 180-dah, which was 4- and 2.6-fold higher than the levels in the telencephalon and diencephalon at 180-dah, respectively (Figure 2E).

### 2.3. Localization of sf-1, dax-1 and star Transcripts in the Brain of the Protogynous Orange-Grouper during Different Stages of Developmental

Three hypothalamic brain areas were selected for the in situ hybridization analysis corresponding to the series Sections 1, 2, and 3, the most enriched expression zone for the genes *star*, *sf-1,* and *dax-1* (Figure 1). The developmental mRNA distribution pattern of *star*, *sf-1,* and *dax-1* in the hypothalamic areas of the grouper brain was demonstrated at 90-, 110-, and 150-dah (Figure 3, Figure 4 and Figure 5). *star*, *sf-1*, and *dax-1* transcripts were found in the ventral habenular nucleus, ventromedial thalamic nucleus, diencephalic ventricle, anteroventral part of the parvocellular preoptic nucleus, gigantocellular part of the magnocellular preoptic nucleus, and lateral tuberal nucleus, dorsal and lateral parts of the hypothalamic nucleus (Figure 3a–u and Figure 5a–u). Localization of *star* (at 110-dah) (Figure 3h–n) and *sf-1* (at 110- and 150-dah) (Figure 4h–u) transcriptional signals were primarily found in cells of the ventral zone of the periventricular hypothalamus. However, *dax-1* transcripts were higher at 90-dah (Figure 5a–g) compared to 110-dah (Figure 5h–n) and 150-dah (Figure 5o–u) in the ventral zone of the periventricular hypothalamus (Figure 5o–u). Parallel brain sections hybridized with the sense probes revealed no positive mRNA signals (Appendix A).

### 2.4. Effect of E_2_ on star, sf-1 and dax-1 mRNA Expression in the Grouper Brain In Vivo

Exogenous E_2_ stimulated *star* expression in all regions of the brain (Figure 6A), but *sf-1* expression was significantly (*p* < 0.05) higher in the mesencephalon and diencephalon but not in the telencephalon, according to the q-PCR results (Figure 6B). In contrast to the telencephalon, E_2_ significantly (*p* < 0.05) reduced *dax-1* expression in the mesencephalon and diencephalon (Figure 6C).

## 3. Discussion

This all-female brain of juvenile orange-spotted grouper (a protogynous species) provides a unique model to study early brain development. We have shown the developmental expression and localization of steroidogenic factors such as *sf-1* and *dax-1* transcripts in the early female brain of the orange-spotted grouper at different developmental ages in comparison to another key steroidogenic enzyme gene (*star*). The peak expression of *sf-1* and *star* genes, as well as those cellular levels and effects of exogenous E_2_ in the early brain, were the most notable findings of this study. In comparison to *sf-1* expression, *dax-1* expression was lower in the early development of the grouper brain. These intriguing findings suggest that *sf-1* may have a regulatory role in steroidogenesis in the early brain of orange-spotted grouper.

*sf-1* is an obvious candidate gene to explain the developmental expression pattern in neural estrogen synthesis in the early female brain, as it regulates the transcription of various target genes involved in steroidogenesis. The expression profiles of *star*, *sf-1,* and *dax-1* are correlated to the time at which most of the other key steroidogenic genes (*cyp11a1*, *hsd3b1*, *cyp17a1,* and *cyp19a1b*) and estrogen receptors (*erα, erβ1, erβ2,* and *gpr30*) exhibited maximal expression in the early female grouper brain [4,6]. In contrast to this scenario, lower expression of *dax-1* was found in the early brain during gonadal sex differentiation when *sf-1* and other key steroidogenic enzymes (including Cyp19a1b activity) were high, indicating that the grouper brain has a functional peak of neurosteroidogenis that may be regulated by *sf-1* and *dax-1* (Figure 7). As a result, at 110-dah, decreased expression of *dax-1* and increased expression of *sf-1*, as well as significantly increased expression of other key steroidogenic genes, may suggest that these orphan nuclear receptor genes play an important role in brain steroidogenesis during the early developmental period.

Until now, available studies in the brain have shown that the expression and localization of orphan nuclear receptors in response to the development of the hypothalamus [19,51], rather than the regulation of brain steroid synthesis during early brain development, as compared to the gonad and interrenal organ [45,46,52]. Our in situ hybridization study revealed that *sf-1* and *star* transcripts were highly expressed at 110- to 150-dah and 110-dah, respectively; while *dax-1* exhibited higher transcripts at 90-dah and a decreasing trend towards 150-dah in cells of the ventral habenular nucleus, ventromedial thalamic nucleus, and ventromedial hypothalamus. In monosex rainbow trout, *Oncorhynchus mykiss*, *sf-1* mRNA expression was localized in the mediobasal hypothalamus [7]. The expression of *SF-1* in the hippocampus of marmosets and rats has been examined using immunohistochemistry [11]. Furthermore, in the marmoset and rat, all StAR-positive cells were also SF-1 positive, demonstrating that the expression of SF-1 and StAR has a functional relationship [11]. SF-1 and DAX-1 immunoreactive cells were discovered in the mouse VMH and pituitary throughout development [23]. Knockout mice deficient in *SF-1* have profound defects in the VMH, strongly suggesting the presence of *SF-1* target genes at this site [51]. Therefore, our findings revealed that the hypothalamus is a key site for the expression and function of *sf-1* and *dax-1*. As a result, the intensive mRNA localization of *star*, *sf-1,* and *dax-1* in the hypothalamic areas of the grouper brain agrees with these previous investigations, indicating that the functional linkage of these genes is plausible for steroidogenesis and brain development.

Furthermore, the current study highlights the in vivo effect of exogenous E_2_ on the regulation of *star*, *sf-1*, and *dax-1* in the grouper early brain. E_2_ is the most biologically prominent and active member of the estrogen family of steroids, and it has a wide range of effects on the developing brain [53]. According to a recent study, the peak expression of numerous neurosteroidogenic genes in the brain of black porgy at the time of gonadal differentiation is mediated by both E_2_-independent and E_2_-dependent pathways [54]. E_2_ exposure was found to up-regulate *star* mRNA in the brain and gonad of the self-fertilizing fish, *Kryptolebias marmoratus* [55]. On the other hand, the xenoestrogen, diethylstilbestrol exposure decreased in the total amount of *sf-1* mRNA in the unborn rat testis but not in fetal ovaries [56]. Holt (1989) [57] proposed the possibility of E_2_, stimulating the transport of cholesterol to the inner mitochondrial membrane. This is consistent with our findings, which reveal that E_2_ exposure increases the expression of *sf-1* and *star* mRNA.

Our in vivo results showed that the exogenous E_2_ up-regulated *star* (telencephalon, mesencephalon, and diencephalon) and *sf-1* (mesencephalon and diencephalon) transcripts compared to their control counterparts. This upregulation could be due to E_2_ binding with nuclear estrogen receptors (*ers*) and *ers*/*sf-1* express in the same cells, or an estrogen-responsive element (ERE) located on the promoter region of the *star* and *sf-1* in the brain. These proposals are unclear, and further research is needed to determine whether the *star*, *sf-1,* and *ers* co-express in the same cells. E_2_ exposure, on the other hand, reduced *dax-1* mRNA levels in the mesencephalon and diencephalon. The overexpression of *sf-1* mRNA and the downregulation of *dax-1* mRNA are strongly related to the results of brain steroidogenesis and E_2_ exposure, owing to the opposing functions of *dax-1* and *sf-1* in steroidogenesis. As a result, for E_2_ upregulation of steroidogenesis in the brain, *sf-1* is more significant than *dax-1*. We previously reported that during early development, E_2_, aromatase enzyme activity, *cyp19a1b*, and estrogen receptors (*er*, *er1,* and *er2*) are all at their greatest levels in the brains of orange-spotted grouper and black porgy [2,3,4,5,6]. Thus, our findings imply that during the early developmental period, there is a local production of neurosteroids, particularly neuroestrogens, that is tightly regulated in the preparation for the peak of functional brain growth (neurogenesis).

Therefore, this current and our previous studies [4,5,6] show the temporal expression and localization of *sf-1* and *dax-1* transcripts in the early brain along with neurosteroidogenic related genes (*star*, *cyp11a1*, *cyp17a1*, and *cyp19a1b*) in the mono-female sex-differentiated fish (Figure 7). *sf-1* transcripts were increased in the early brain as other key steroidogenic genes and E_2_ [2,3,4,58]; however, *dax-1* had an antagonistic expression pattern during these time periods, suggesting that *dax-1* may play a negative role in the early brain development of female grouper. Indeed, the *star*, *sf-1,* and *dax-1* mRNAs were found in abundance in the cells of the ventromedial thalamic nucleus and the mediobasal hypothalamus, and these three genes were mostly found in the same regions, implying that their functions in steroidogenesis during brain development are closely related. Furthermore, exogenous E_2_ also increased the expression of *star*, *cyp19a1b*, and *sf-1* in the brain, indicating that these genes are primarily involved in E_2_ synthesis. This finding led to a better understanding of the transcriptional regulation of *sf-1* and *dax-1* as molecular players for steroidogenesis during development (Figure 7). Therefore, the balance of *sf-1* and *dax-1* expression is critical in the regulation of brain development and sex differentiation.

## 4. Materials & Methods

### 4.1. Experimental Fish

Orange-spotted female grouper (*E. coioides*) were used in the present study. The experimental fish were acclimated to the pond condition of the University culture station in a seawater and natural light system (salinity of 33 ppt; temperatures ranged from 20–24 °C). The fish were fed *ad libitum* with a commercial food (Fwu Sou Feed Co., Taichung, Taiwan). All procedures and investigations were approved by the National Taiwan Ocean University Institutional Animal Care and Use Committee and were performed in accordance with standard guiding principles.

Three batches of orange-spotted groupers were used for the experiment. For the gene expression analysis, fish (*n* = 8 per age group) were obtained at 90-, 110-, 130-, 150-, and 180-dah. To investigate the localization of genes in hypothalamic brain regions, fish with 90-, 110-, and 150-dah were obtained. To investigate the effects of exogenous E_2_ on gene expression, 110-dah fish were obtained. The fish were anesthetized with 0.05% ethylene glycol monophenyl ether before being decapitated. The brain was dissected in the same manner as previously described for the orange-spotted grouper and black porgy [4,6,54,59,60]. Brain samples, including telencephalon (including the olfactory bulb, telencephalon, and part of the preoptic area; located between the anterior commissure and the optic chiasm), mesencephalon, and diencephalon (including mesencephalon, thalamus, epithalamus, subthalamus, and hypothalamus), were immediately frozen in liquid nitrogen and stored at −80 °C until RNA isolation. Another batch of whole-brain tissues was fixed in a 4% paraformaldehyde solution for histological/cellular analysis.

#### 4.1.1. Experiment 1: Gene Expression Profiles during Gonadal Sex Differentiation

In order to examine the gene expression profiles in the brain during different developmental ages and gonadal sex differentiation, brain samples (*n* = 8) were collected from 90-dah (body weight, BW = 11.6 ± 0.5 g, body length, BL = 9.1 ± 0.2 cm), 110-dah (BW = 24.5 ± 0.8 g, BL = 11.7 ± 0.1 cm), 130-dah (BW = 28.1 ± 1.1 g, BL = 12.3 ± 0.2 cm), 150-dah fish (BW = 31.2 ± 1.2 g, BL = 12.9 ± 0.2 cm), and 180-dah (BW = 48.1 ± 2.8 g, BL = 15.3 ± 0.3 cm). The mRNA expression of the genes was measured using q-PCR.

#### 4.1.2. Experiment 2: Localization of *sf-1,*
*dax-1* and *Star* Gene Expression in the Hypothalamus of Orange-Spotted Grouper

We used in situ hybridization in distinct hypothalamic brain regions to determine the anatomical localization of cells that express the *star*, *sf-1,* and *dax-1* genes (see detailed methods below).

#### 4.1.3. Experiment 3: In Vivo Effects of E_2_ on the Expression of mRNAs Encoding *sf-1*, *dax-1,* and *Star* in the Brain of Grouper

In order to further investigate the effects of E_2_ on the mRNAs expression of *sf-1*, *dax-1* and *star*, 110-dah female grouper for each group (BW = 32.3 ± 1.8 g, BL = 13.5 ± 0.25 cm) were divided into two groups, with *n* = 8 fish in each of the following experimental groups: control (vehicle alone, coconut oil; Sigma, St. Louis, MO, USA) and E_2_ treatment (1 µg/g BW; Sigma). On day 1 and 5, the fish received an intramuscular injection (two injections in total). Fish telencephalon, mesencephalon, and diencephalon were collected 24 h (day 6) after the 2nd injection (day 5) and stored at −80 °C until further use. Q-PCR was used to examine changes in the mRNA expression of the corresponding genes in the brains of control and E_2_-treated groups.

### 4.2. RNA Extraction, First-Strand cDNA and Molecular Cloning of Genes in the Grouper Brain

Total RNA was extracted from the telencephalon, mesencephalon, and diencephalon (90- to 180-dah) using the TRIzol reagent method (Gibco BRL; Grand Island, NY, USA) according to the manufacturer’s instructions. The quality and concentrations of RNA were assessed by spectrophotometry and checked by running an aliquot (1 μg) on a 1.8% agarose-formaldehyde gel. Single-stranded cDNA was constructed from total RNA using Invitrogen reagents (Invitrogen, Carlsbad, CA, USA), oligo (DT) 12–18 primers and Superscript II reverse transcriptase (Gibco BRL) in a 20 µL reaction volume with an incubation at 42 °C for 60 min, 37 °C for 15 min, and 70 °C for 15 min. The resulting cDNAs were used as a template for subsequent PCR amplification. The PCR reaction was performed in a final volume of 25 μL reaction containing 2.5 μL of 10× reaction buffer (200 mM Tris–HCl, pH 8.4, 500 mM KCl), 1μL of 10 mM dNTP, 1 μL of 2 mM MgCl_2_, 0.5 μL each of 10 μM forward and reverse primer, respectively (Mission Biotech Co., Ltd., Taipei, Taiwan), 0.2 μL superscript enzyme (Invitrogen, Carlsbad, CA, USA), and 1 μL cDNA. The reaction conditions for degenerate PCR were as follows: 94 °C for 5 min, 35 cycles of 94 °C for 30 s, 50 °C for 30 s, 72 °C for 30 s and 72 °C for 7 min. Each PCR product was electrophoresed on 1.5% agarose gel and the fragment showing the predicted molecular weight was then excised using Gel-MTM Gel Extraction system kit (Viogene, Bio 101, La Jolla, CA, USA). Extracted cDNAs were ligated into the pGEM-T Easy vector (Promega, Madison, WI, USA) and transformed into *Escherichia coli* competent cells following the manufacturer’s instruction. The Plasmid containing the insert was sequenced and compared with the NCBI database using BLAST. The genes *star* (GenBank accession no. GU929702), *sf-1* (JQ320496), and *dax-1* (GU929703) were cloned from the grouper brain, which was deduced from the conserved regions of other teleosts (Appendix A).

### 4.3. Preparation of Brain Tissue for In Situ Hybridization

For in situ hybridization experiments, 90-, 110-, and 150-old female grouper brains were collected to define *star*, *sf-1*, and *dax-1* mRNA signals in the brain. For in situ hybridization, we chose three hypothalamic areas with the most enriched expression zones for the genes *star*, *sf-1,* and *dax-1* (Figure 1). Groupers were anesthetized by immersion in ethylene glycol monophenyl ether (0.05%). Grouper brains were fixed overnight in a 4% paraformaldehyde in PBS (phosphate-buffered saline) at 4 °C. Upon fixation, the skull was removed and the brain was carefully dissected out and stored overnight in fresh fixative at 4 °C. The brain was then rinsed several times in PBS. Later, the fixed brain was dehydrated in a series of alcohols, embedded in paraffin, and then cross-sections (5–6 µm) were collected on APTES-treated slides (3-aminopropyltriethoxysilane, Sigma) diluted in acetone. 

### 4.4. Synthesis of Star, sf-1 and dax-1 RNA Probes

The PCR product of the target gene from the plasmid DNA containing respective inserts of the genes in the pGEM-T Easy vector was generated with 50 U DNA polymerase (New England Biolabs, Ipswich, MA, USA) for DNA amplification with in situ hybridization primers (Table 1). The PCR products were purified using a kit (PCR-Advanced Clean-Up Kit, Viogene). In vitro transcription was carried out using this purified DNA as a template. Digoxigenin-labeled sense and anti-sense riboprobes were synthesized from a fragment of 751 bp, 570 bp, and 443 bp encoding orange-spotted grouper brain *star*, *sf-1*, and *dax-1*, respectively, by using T7 and SP6 polymerase (Promega). The DNA (1µg) templates were incubated for 3 h at 37 °C in the thermocycler PCR machine (Applied Biosystems, Foster City, CA, USA) in a solution containing transcription buffer (5X), 0.1 M dithiothreitol (DTT), DIG-rNTP mix (10X) (Roche, Penzberg, Germany), an RNase inhibitor (40 U/µL) (Promega) and T7 or SP6 RNA polymerase (20 U/µL) and adjusted to 20 µL final volume with sterile DEPC H_2_O. The extra template was removed by digesting it with 4 µL of DNase I (10 U/µL) at 37 °C for 30 min. After incubation, the probes were precipitated with 100 µL of LiCl (7.5 M) and 900 µL of isopropanal at −20 °C overnight. The pellets were collected by centrifuging the solution at 12,000 rpm at 4 °C for 30 min and the pellets were re-suspended with 2 µL RNase inhibitor and 98 µL sterile DEPC H_2_O. The probe quality was checked by spectrophotometry at 260 nm.

### 4.5. In Situ Hybridization

The in situ hybridization analysis was performed as described in our previous study [6,59,60,61,62,63]. The adjacent sections were rehydrated from xylene to a series of alcohol (100%, 90%, 80%, 70%, and 50% EtOH) and washed in PBS several times. Then, they were treated with proteinase K at 10 µg/mL (Roche, Mannheim, Germany) at room temperature for 8 or 10 min and rinsed in PBST (Tween-20 in phosphate-buffered saline) and PBS several times. The brain sections were then prehybridized in the prehybridization buffer for 1 h at 68 °C and hybridized with the sense or anti-sense digoxigenin-labeled RNA probes (0.5 µg/mL for *sf-1* and *dax-1*, and 1 µg/mL for *star*) in 200 µL hybridization buffer containing 50% formamide, 5X SSC, 500 µg/mL tRNA, 50 µg/mL heparin, and 0.1% Tween-20 (pH 6.0) at 68 °C overnight. After overnight incubation, the sections were allowed to room temperature for 10 min, rinsed with buffer containing 25% formamide, 1X SCC, and 0.1% Tween-20 at 60 °C for 15 min two times, and blocked with 2% blocking reagent (Roche), 2% normal goat serum and PBT at room temperature for 1 h. Alkaline phosphatase-conjugated sheep anti-digoxigenin antibody (Roche) (dilution 1:1500 at 2% blocking reagent) was applied in the section and then incubated overnight at 4 °C. After washing, NTMT (100 mM NaCl, 100 mM Tris-HCl, pH 9.5, 50 mM Mg_2_Cl, 0.1% Tween-20) was used to ensure the pH is high. Finally, the NBT/BCIP (nitro blue tetrazolium/5-bromo-4-chloro-indolyl-phosphate) staining (Sigma) was used to detect the mRNA signal of grouper steroidogenic genes in the brain.

### 4.6. Q-PCR Analysis

Transcripts of grouper *star* [4], *sf-1,* and *dax-1* were quantified by quantitative real-time PCR analysis in the telencephalon, mesencephalon, and diencephalon at different developmental ages (90- to 180-dah). Q-PCR primers were designed for the analysis of *star*, *sf-1*, *dax-1,* and *β-actin* with the assistance of Primer Express Software (Applied Biosystems) (Table 1). The expression levels of brain *β-actin* were not significantly different among development ages (Appendix A). *β-actin* was used as a housekeeping gene in the q-PCR analysis to calibrate the expression levels of brain genes. Gene quantification of the standards, samples, and control was simultaneously conducted in a q-PCR machine (iQTM Multicolor Real-Time PCR Detection System; Bio-Rad Co., Hercules, CA, USA) with iQTM SYBR green (Bio-Rad Co.), which is a dsDNA minor-groove binding dye, forward and reverse primers and water for the reaction mix in a MicroAmp^®^ 96-well reaction plate. The respective standard curve of the log (transcript concentrations) vs. CT (the calculated fractional cycle number at which the PCR-fluorescence product is detectable above a threshold) was −0.995.

### 4.7. Statistical Analysis

All values (*n* = 8) are represented as the mean ± SEM (standard error of mean), and the data were analyzed by one-way ANOVA followed by an S–N–K (Student–Newman–Keuls) multiple comparison test to compare the differences (*p* < 0.05) in the developmental expression of the genes in different parts of the brain at different ages. Student *t*-test was also conducted to compare the significant differences (*p* < 0.05) between control and E_2_-injected fish.

## 5. Conclusions

In conclusion, the current study found developmental changes in the mRNA expression and localization of orphan nuclear receptors such as *sf-1* and *dax-1* in relation to neurosteroidogenic enzyme gene in the early brain of the orange-spotted grouper. It has been suggested that *sf-1* and *dax-1* play an important role with key enzymes in steroidogenesis in the early brain of the orange-spotted grouper. Furthermore, our findings revealed that E_2_ increased *star* and *sf-1* mRNA expression in the telencephalon and diencephalon, which is critical for early brain development and steroidogenesis (Figure 7). Overall, this research found that the orange-spotted grouper brain has all the necessary mechanisms for neurosteroidogenesis and, as a result, local E_2_ synthesis, which may regulate early brain development during gonadal sex differentiation.

## Figures and Tables

**Figure 1 ijms-23-02614-f001:**
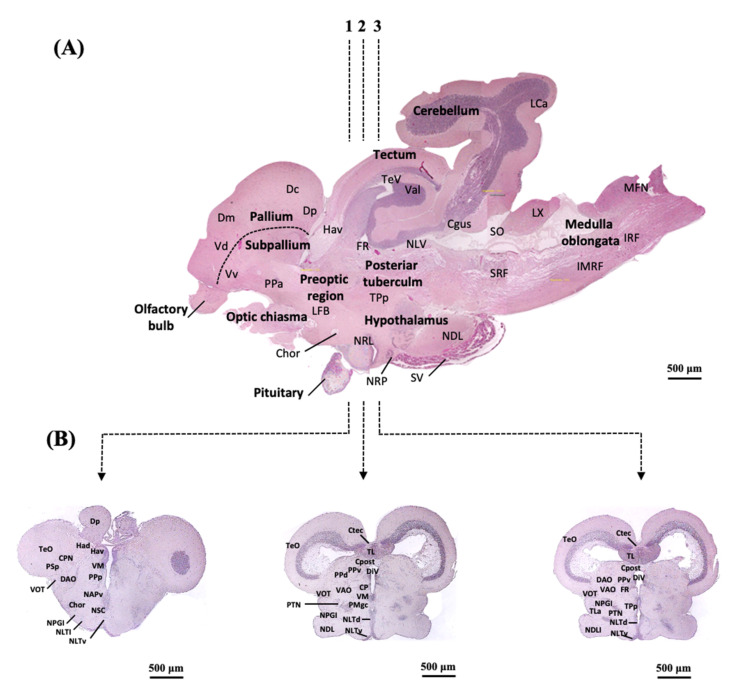
Neuroanatomy section (**A**) the whole brain of the orange-spotted grouper showing different areas of the olfactory bulb, telencephalon, diencephalon, mesencephalon, rhombencephalon and (**B**) showing three chosen areas of the brain for the in situ hybridization correspond to the series Sections 1, 2, and 3 where most enriched expression zone for the genes *star*, *sf-1,* and *dax-1*. Dashed lines showed the dissection position of transverse section. Hematoxylin and eosin were used to stain brain sections. aNH, anterior neurohypophysis; Chor, commissura horizontalis; Cgus, commissure of the secondary gustatory nuclei; CPN, central pretectal nucleus; Ctec, commissural tecti; Cpost, posterior commis-sure; DAO, dorsal accessory optic nucleus; Dc, central zone of dorsal telencephalic area; DiV, diencephalic ventricle; Dm, medial zone of dorsal telencephalic area; Dp, posterior zone of dorsal telencephalic area; FR, fasiculus retroflexus; Hav, ventral habenular nucleus; IMRF, intermediate reticular hypothalamus; IRF, inferior reticular formation; LCa, lobus caudalis cerebella; LFB, lateral forbrain bundle; LX, vagal lobe; MFN, medial funicular nucleus; NDL, diffuse nucleus of the inferior lobe; NLV, nucleus lateralis nucleus; NPGl, lateral preglomerular nucleus; NRL, nucleus of the lateral recess; NRP, nucleus of the posterior recess; NLTd, lateral tuberal nucleus, dorsal part; NLTv, lateral tuberal nucleus, ventral part; PI, pars intermedia; PMgc, gigantocellular part of the magnocellular preoptic nucleus; pNH, posterior neurohypophysis; PPa, parvocellular preoptic nucleus, anterior part; PPd, proximal pars distalis; PPv, periventricular pretectal nucleus, ventral part; PSp, parvocellularsuperficialpretectalnucleus; PTN, posteriortuberalnu-cleus; SO, secondary octaval population; SRF, superior reticular formation; SV, saccus vasculosus; TeV, tectal ventricle; TeO, tectum opticum; TL, torus longitudinalis; TPp, periventricular nucleus of the posterior tuberculum; Val, lateral division of the valvula cerebella; VM, ventromedial thalamic nucleus; VMH, ventromedial hypothalamus.

**Figure 2 ijms-23-02614-f002:**
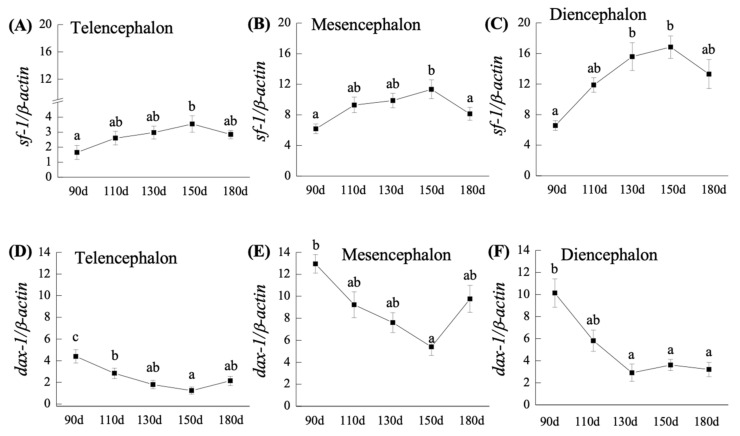
Relative transcripts of *sf-1* (**A**–**C**) and *dax-1* (**D**–**F**) in the telencephalon, mesencephalon, and diencephalon from 90- to 180-dah (days after hatching) during gonadal sex differentiation. Gene expression levels were measured by q-PCR and expressed as mean normalized expression (SEM) of *n* = 8 fish for each group. Different letters denote statistically significant differences (*p* < 0.05) in gene expression at various developmental ages.

**Figure 3 ijms-23-02614-f003:**
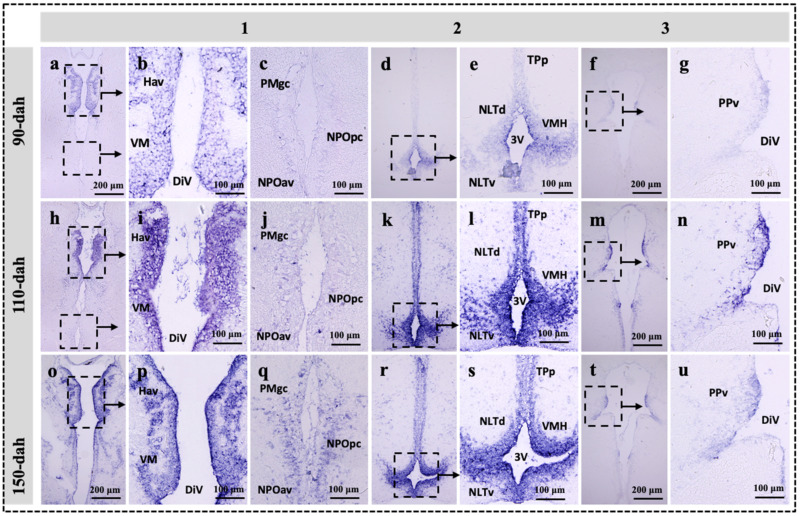
Developmental localization of *star* transcripts in the hypothalamic regions of the grouper brain at 90-, 110-, and 150-dah (day after hatching). Bright field visualization of paraffin brain sections showing the detailed distribution of *star* transcripts in three consecutive area of the hypothalamus at 90- (**a**–**g**), 110- (**h**–**n**), and 150-dah (**o**–**u**) corresponding to the series 1, 2 and 3 on Figure 1. Note that intense *star* hybridization signals were detected at 110-dah in the ventromeadial thalamic region (**h**,**i**), ventromeadial hypothalamic region (**k**,**l**), and periventricular pretectal regions (**m**,**n**) compared to the same region at 90 (**a**–**g**) or 150-dah (**o**–**u**). The abbreviations are referred to Figure 1 legends.

**Figure 4 ijms-23-02614-f004:**
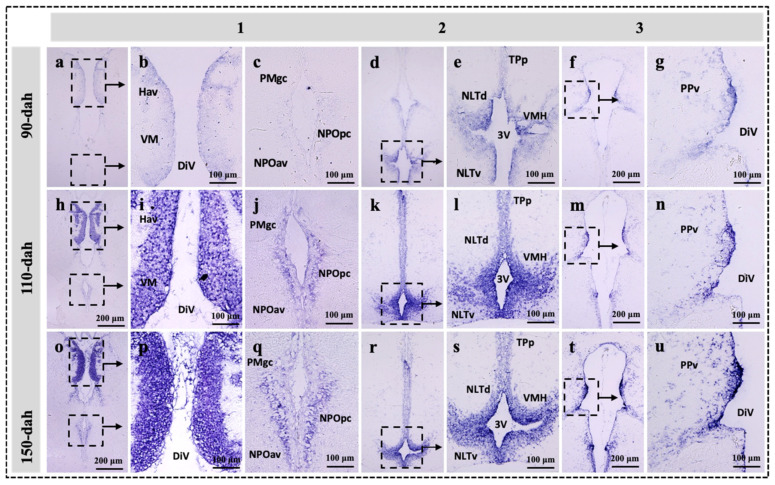
Developmental localization of *sf-1* transcripts in the hypothalamic regions of the grouper brain at 90-, 110-, and 150-dah (day after hatching). Bright field visualization of paraffin brain sections showing the detailed distribution of *sf-1* transcripts in three consecutive area of the hypothalamus at 90- (**a**–**g**), 110- (**h**–**n**), and 150-dah (**o**–**u**) corresponding to the series 1, 2 and 3 on Figure 1. Note that intense *sf-1* hybridization signals were detected in 110-dah in the ventromeadial thalamic region (**h**,**i**), ventromeadial hypothalamic region (**k**,**l**), and periventricular pretectal regions (**m**,**n**) compared to the same region at 90 (**a**–**g**) or 150-dah (**o**–**u**). The abbreviations are referred to Figure 1 legends.

**Figure 5 ijms-23-02614-f005:**
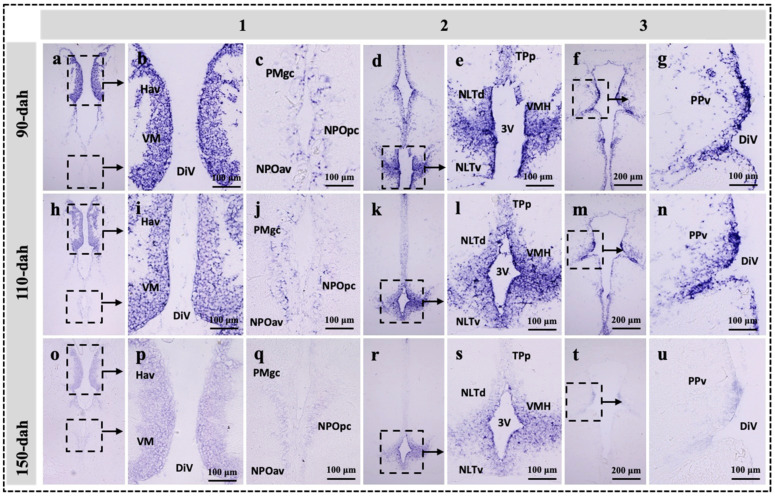
Developmental localization of *dax-1* transcripts in the hypothalamic regions of the grouper brain at 90-, 110-, and 150-dah (day after hatching). Bright field visualization of paraffin brain sections showing the detailed distribution of *dax-1* transcripts in three consecutive area of the hypothalamus at 90- (**a**–**g**), 110- (**h**–**n**), and 150-dah (**o**–**u**) corresponding to the series 1, 2 and 3 on Figure 1. Note that intense *dax-1* hybridization signals were detected in 90-dah in the ventromeadial thalamic region (**a**–**c**), ventromeadial hypothalamic region (**d**,**e**), and periventricular pretectal regions (**f**,**g**) compared to the same region at 150-dah (**o**–**u**). The abbreviations are referred to Figure 1 legends.

**Figure 6 ijms-23-02614-f006:**
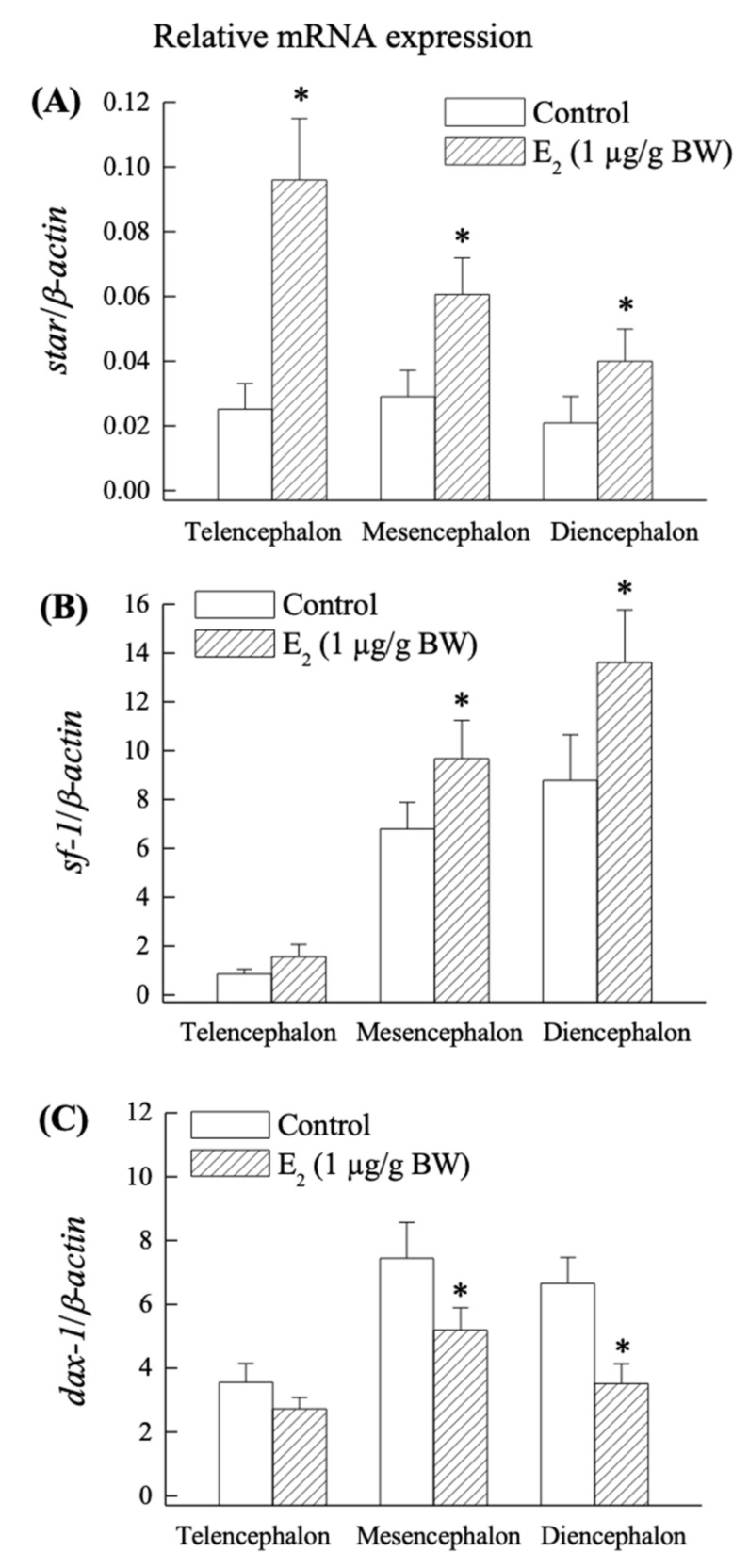
In vivo effects of exogenous estradiol (E_2_, 1 µg/g BW, fish *n* = 8 (110-dah, day after hatching) for each group were given intramuscular injections on day 1 and 5). Brain tissues were collected on day 6 for the analysis of mRNA expression of *star* (**A**), *sf-1* (**B**), and *dax-1* (**C**), as measured by q-PCR analysis in the telencephalon, mesencephalon, and diencephalon. The results are expressed as the mean with standard deviation (SEM). The different letters indicate significant differences (*p* < 0.05) according to a one-way ANOVA followed by a S–N–K (Student–Newman–Keuls) multiple comparison test. An asterisk (*) represents significant (*p* < 0.05) differences between the control and E_2_-injected groups.

**Figure 7 ijms-23-02614-f007:**
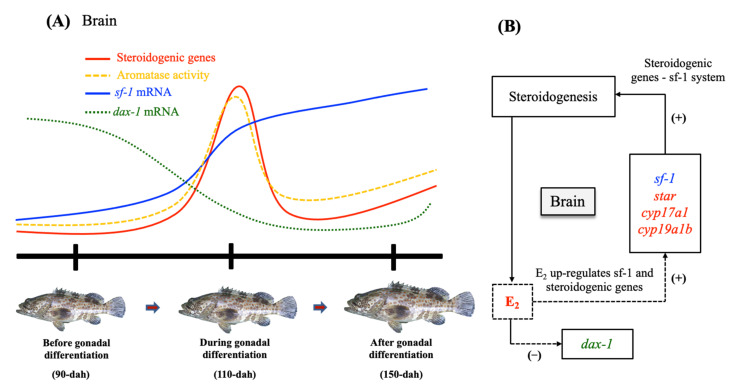
Schematic picture highlighting the proposed hypothesis according to the current and previous studies [4,5,6]. (**A**) The expression profiles of steroidogenic enzyme genes (*star*, *cyp17a1*, and *cyp19a1b*) [4,5,6], aromatase activity [5], *sf-1* and *dax-1*. (**B**) Brain estradiol (E_2_) can act locally in the regulation of steroidogenic enzyme genes [6] and *sf-1*; and collectively E_2_-*sf-1*-steroidogenic enzymes system may stimulate steroidogenesis, neurogenic activity, and early brain development in the juvenile orange-spotted grouper. (+) stimulation and (−) inhibition.

**Table 1 ijms-23-02614-t001:** Primer sequence used for q-PCR and in situ hybridization analysis for the genes *star*, *sf-1,* and *dax**-1* in the orange-spotted grouper. The nucleotide (nt) length of RNA probes for in situ hybridization: *star,* 751 nt; *sf-1,* 570 nt; *dax-1*, 443 nt.

Gene	Orientation	Nucleotide Sequence	Usage	Accession Number
*star*	Forward	5′-TCAGCACAGGGCTTCATCACTAT-3′	q-PCR	
	Reverse	5′-TGCAAAAATGCCTGAGCAAAG-3′	q-PCR	
	Sense	5′-CGGCATATGAGGAACATGACAG-3′	In-situ	GU929702
	Anti Sense	5′-CCACCTGCGTCTGAGAGAG-3′	In-situ	
*sf-1*	Forward	5′-TCGGCTTCGTGAAAAACGT-3′	q-PCR	
	Reverse	5′-CTCCTGGGCATGGCTCAA-3′	q-PCR	
	Sense	5′-GCTGCAAGGGGTTCTTCAAG-3′	In-situ	JQ320496
	Anti Sense	5′-CGGGGTACTCTGACTTGATG-3′	In-situ	
*dax-1*	Forward	5′-TGAAGAAGTGCTGGAGTGTAGATATCA-3′	q-PCR	
	Reverse	5′-CCCTCCACATCTGGGTTGAA-3′	q-PCR	
	Sense	5′-CTCGGCGGTGCTGGTGAAGAC-3′	In-situ	GU929703
	Anti Sense	5′-GCCGGACGTGCTCGTTGAGAG-3′	In-situ	
*β-actin*	Forward	5′-AGGGAGAAGATGACCCAGATCA-3′	q-PCR	AY510710
	Reverse	5′-GGGACAGCACAGCCTGCAT-3′	q-PCR	

## Data Availability

Data are contained within this article. Raw data are available on request from the corresponding authors.

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
