# Peer review of "Expression and Transcript Localization of star, sf-1, and dax-1 in the Early Brain of the Orange-Spotted Grouper Epinephelus coioides"

_ijms, 2022, doi:10.3390/ijms23052614_

Round 1

Author Response

Thank you very much for your detailed, intelligent, and insightful comments on our paper, which helped us to improve it. We had carefully made changes (highlighted with tracked changes) according to the your comments. We had modified Figures 1 - Figure 7, added one new Table (Table 1) and five supplementary figures (Figure S1-S5). 

Reviewer 2 Report

The authors performed appropriate experiments to conclusions. The conclusion is clear, and it will have a great impact on the scientific community.
Manuscript must be corrected with inappropriate figure layout and writing style. With inadequate writing of good findings, the value of the findings diminished.

  1. Forms are not unified at all. For example, lines 13-15 font size, italics of "sf-1" and a mix of uppercase and lowercase letters. Proofreading the English language and formatting should be done carefully. Since "in situ" in lines 331 and 383 is derived from Latin, it must be expressed in italics. Line 358 is well expressed. Line 78 "cis" also must be expressed in italics.
  2. Abstract: It is well written to show the overall content well. However, the last sentence of line 29 is awkward. Since the results of the study have already been presented, it should be removed.
  3. Introduction: This paper is a research article, not a review paper. Overall, the introduction section is too long. It should be deleted, leaving only the important points. However, species-specific expression, roles, or evolutionary patterns of sf-1 and dax-1 are important for future studies. It would be good to organize them in figures (especially lines 65-69, 77-79) and introduce them as review papers later.
  4. Introduction: "P450scc" (Line 45), "P450arom" (Lines 59, 85): Are these terms used in academia? It would be better to express it as CYP11A1, etc.
  5. Results: Figures 1, 3, 4 and 5 are anatomical schematic diagrams and although they are important figures, it is recommended to use them as supplementary figures. It should be presented as a high-resolution figure.
  6. Figures: A "figure" letter is indicated in the lower left corner of every figure. It should be deleted.
  7. Figure 6: Please provide full-name (Tel -> Telencephalon, etc). Please change vertical layout to horizontal.
  8. Figure 7: This figure is an important figure that presents the core topic of the paper. Overall layout and color selection are good. The light blue color of aromatase activity and the blue color of sf-1 mRNA are not well distinguished. You will have to distinguish them by other colors (such as orange). It would be nice to display E2 in a different color. "A)" -> "(A), "B)" -> "(B)".
  9. Discussion: The Discussion section is too long. It should be written mainly on important content (concordance between the results of this study and the results of previous studies) and omitted.
  10. Method: If the pipeline of this study is used for follow-up studies, the method should be written in detail. In the paper of subsequent research, the paper should be cited and briefly written.
  11. Method: Lines 411-415, provided as table.
  12. Table 1: It's not organized at all. It need arrange.

Author Response

Thank you very much for your valuable feedback on our paper, which helped us to improve it. We had carefully made changes (highlighted with tracked changes) according to the your comments. We had modified Figures 1 - Figure 7, added one new Table (Table 1) and five supplementary figures (Figure S1-S5). 

Round 2

Reviewer 1 Report

Thanks for your reply with great work done for the revision. I am very glad to see that the revised manuscript has addressed all my questions clearly. It apparently shows your passion and great efforts had been done on the grouper research. The novel findings reported here are promising and interesting.  I can not do more than suggest it for the acceptance now.

After reading the paper again, I did not find out any place with mistakes. The supplementary figs look good. But you may need to do one or two more rounds of proof reading of the text including those fig legends to get rid of typos. Now I can understand Fig7 in the conclusion easily. Please make sure that 90-dah is only a little smaller than 150-dah without any other morphological changes. I know nothing about it. Hope you are right.  

Reviewer 2 Report

The authors modified the paper well in line with the reviewer's point. I hope that this paper will be of great help in the scientific field. Please check minor typo errors in the clean version before publication.